# Harnessing Carcinoma Cell Plasticity Mediated by TGF-β Signaling

**DOI:** 10.3390/cancers13143397

**Published:** 2021-07-07

**Authors:** Xuecong Wang, Jean Paul Thiery

**Affiliations:** Bioland Laboratory, Guangzhou Regenerative Medicine and Health, Huangpu District, Guangzhou 510700, China

**Keywords:** epithelial cell plasticity, epithelial–mesenchymal transition, TGF-β, chemoresistance, stemness, invasion, metastasis

## Abstract

**Simple Summary:**

This review describes mechanisms driving epithelial plasticity in carcinoma mediated by transforming growth factor beta (TGF-β) signaling. Plasticity in carcinoma is frequently induced through epithelial–mesenchymal transition (EMT), an evolutionary conserved process in the development of multicellular organisms. The review explores the multifaceted functions of EMT, particularly focusing on the intermediate stages, which provide more adaptive responses of carcinoma cells in their microenvironment. The review critically considers how different intermediate or hybrid EMT stages confer carcinoma cells with stemness, refractoriness to therapies, and ability to execute all steps of the metastatic cascade. Finally, the review provides examples of therapeutic interventions based on the EMT concept.

**Abstract:**

Epithelial cell plasticity, a hallmark of carcinoma progression, results in local and distant cancer dissemination. Carcinoma cell plasticity can be achieved through epithelial–mesenchymal transition (EMT), with cells positioned seemingly indiscriminately across the spectrum of EMT phenotypes. Different degrees of plasticity are achieved by transcriptional regulation and feedback-loops, which confer carcinoma cells with unique properties of tumor propagation and therapy resistance. Decoding the molecular and cellular basis of EMT in carcinoma should enable the discovery of new therapeutic strategies against cancer. In this review, we discuss the different attributes of plasticity in carcinoma and highlight the role of the canonical TGFβ receptor signaling pathway in the acquisition of plasticity. We emphasize the potential stochasticity of stemness in carcinoma in relation to plasticity and provide data from recent clinical trials that seek to target plasticity.

## 1. Introduction

Cell plasticity can be defined as the ability of a cell to modify its phenotype in response to its microenvironment. In carcinoma, plasticity is an exquisite adaptive mechanism to respond to stressors in the microenvironment, endowing cells not only with the capacity to avoid apoptosis, to acquire stemness and refractoriness to therapy and to transdifferentiate into other cell types. Understanding cell plasticity in carcinoma should help to decipher the intricate mechanisms driving tumor propagation and drug resistance. Plasticity is primarily conferred by epithelial–mesenchymal transition (EMT), an intrinsic function in the construction of the body plan in metazoans that becomes jeopardized during carcinogenesis. EMT is a complex process driven by epigenetic and transcriptional regulators. EMT is also highly dynamic and partially reversible, thus positioning carcinoma in an N-dimensional diagram reduced to three dimensions; this is often provided as a Waddington-type diagram to emphasize the epigenetic and energetic intermediate states within the EMT spectrum [1]. A deeper understanding of the cellular and molecular mechanisms driving EMT and how these endow cells with plasticity is essential for the design of new therapeutic strategies for patients with cancer.

In this review, we highlight the recent findings relevant to epithelial cell plasticity in carcinoma and discuss the transcriptional regulators and the impact of transforming growth factor beta (TGF-β) signaling on this pathway. We also question the extent to which EMT is involved in the acquisition of stemness and therapeutic resistance. Finally, we describe some new therapeutic interventions relevant to the plasticity phenotype of carcinoma.

## 2. Transcriptional Regulation of Carcinoma Plasticity

Epithelial cell plasticity describes how epithelial cells are able to alter their morphology and fate through distinct epigenetic and transcriptional regulatory mechanisms. Although not discussed in this review, mechanotransduction also plays a major role in the execution of plasticity [2]. In epithelial cells, plasticity may be associated with reduced proliferation along with adaptations to the microenvironment, which in turn, may modify their identity in normal tissues. Epithelial cell plasticity also operates during carcinoma progression [3,4]. One of the key mechanisms driving carcinoma cell plasticity is epithelial–mesenchymal transition (EMT): a fundamental, dynamic process that allows epithelial cells to lose their apico-basal polarity and acquire front-rear polarity in the transition toward a mesenchymal-like phenotype. EMT is essential in embryonic development particularly during gastrulation, in neural crest ontogeny, and in heart morphogenesis [5]. EMT also contributes to wound-healing [6] and indirectly to tissue fibrosis [7,8].

EMT is a highly dynamic and extremely complex process, controlled by numerous regulatory networks. EMT signaling networks lead to a repression of cell–cell adhesion, a loss of epithelial cell polarity and a gain of mesenchymal-like migratory phenotypes. Specifically, there are several EMT-inducing transcription factors routinely identified in studies as key factors driving the EMT response and leading carcinoma plasticity, some of which will be explored here further.

Zinc finger E-box binding homeobox 1 (ZEB1) and zinc finger E-box binding homeobox 2 (ZEB2) proteins can repress not only the transcription of the E-cadherin gene [9], but also genes encoding tight junction components, such as claudin-4 and zonula occludens protein 3 (ZO-3), and desmosomal plakophilin-2 gene [10]. In squamous cell carcinoma (SCC) cells, thyroid hormone (TH) works with its activating enzyme D2 to upregulate ZEB1 expression resulting in the induction of EMT [11]. ZEB1 and miR-200 form a double-negative feedback-loop to regulate E-cadherin expression in epithelial and mesenchymal states. This mechanism allows cells to execute a rapid response to stimuli within a short exposure time. However, such a mechanism—related to hysteresis in solids—cannot be activated following deletion of ZEB1 binding sites on the E-cadherin promoter [12]; hysteresis in EMT suggests a state of memory that allows for a strong cellular response, one that causes carcinoma plasticity to persist even after the stimulus is withdrawn. ZEB1 can switch from a repressor into a transcriptional activator in conjunction with the Hippo pathway effector Yes-associated protein (YAP), promoting the transcription of a subset of YAP target genes, such as AXL receptor tyrosine kinase (*AXL*), cyclin-dependent kinase inhibitor 2C (*CDKN2C*), connective tissue growth factor (*CTGF*), cysteine-rich protein 61 (*Cyr61*) and DAB adaptor protein 2 (*DAB2*), which drive carcinoma progression and metastasis [13]. ZEB1 can form a transactivation complex with activating protein-1 (AP-1) and YAP factors directly to promote EMT and tumor progression in triple-negative breast carcinoma [14].

Snail family transcriptional repressor 1 (SNAI1) and Snail family transcriptional repressor 2 (SNAI2) can also bind to E-boxes through their carboxy-terminal zinc-finger domain to repress epithelial gene expression [15]. However, acetylated SNAI1 can act as an activator to induce tumor necrosis factor alpha (*TNF-**α*), C-C motif chemokine ligand 2 (*CCL2*), and C-C motif chemokine ligand 5 (*CCL5*) gene transcription, which in turn promote carcinoma progression and the recruitment of tumor-associated macrophages (TAMs) [16]. In breast carcinoma cells, EMT can be inhibited by SNAI1 degradation through phosphorylation, mediated by the PAR-atypical protein kinase C (aPKC) complex [17]. SNAI2 upregulation can also enhance cell plasticity to promote the invasion and metastasis of squamous cell carcinoma (SCC) [18]. In ovarian cancer, upregulation of programmed cell death 10 (PDCD10) and silencing miR-222-3p microRNA promote carcinoma plasticity, which is mediated by SNAI2 [19].

Twist family bHLH transcription factor 1 (TWIST1) and twist family bHLH transcription factor 2 (TWIST2) are members of the basic helix-loop-helix (bHLH) transcription family, which are key regulators of lineage specification and EMT. In carcinoma, TWIST can repress the expression of epithelial-related genes like *E-cadherin*, and enhances the expression of mesenchymal-related genes like *N-cadherin* and *vimentin* in order to induce cell plasticity [20]. EMT mediated by overexpression of TWIST is associated with hepatocellular carcinoma (HCC) cell invasion and metastasis, possibly through the mediation of Cullin 2 (Cul2) circular RNA to increase the expression of vimentin [21,22]. Overexpression of TWIST1 induces dissemination of cells from breast cancer organoids, a necessary step in metastasis. This process is driven by protein kinase D1 (Prkd1) inactivating β-catenin through serine 33 phosphorylation thus disrupting cadherin-mediated intercellular junctions [23].

In addition to the EMT-inducing transcription factors described above, recent studies have focused on chromatin remodeling as a key factor in the control of plasticity. Chromatin remodeling is a complex process regulating the degree of compaction and thus the accessibility of coding genes and other DNA sequences by regulatory transcriptional protein machinery. Histone modifications by specific enzymes, such as histone acetyltransferases (HATs), deacetylases, and methyltransferases, are important in chromatin remodeling. Protein arginine methyltransferase 5 (PRMT5) complexed with methylosome protein 50 (MEP50)/WD repeat domain 77 (WDR77) can catalyze H3 and H4 arginine methylation, leading to the activation of EMT markers and carcinoma cell metastasis [24]. Lysine-specific demethylase 1 (LSD1) is also essential for EMT and tumor progression, and is highly expressed in human carcinoma cells. When cells are in the epithelial state, LSD1 can be acetylated directly by the males-absent on the first (MOF) acetyltransferase, dissociating LSD1 from nucleosomes and inducing its demethylation function on lysine 4 on histone H3 (H3K4) [25]. MOF acetylation of LSD1 also causes a decrease in its binding to the promoter regions of epithelial genes, such as *E-cadherin* and *KRT8*, thus promoting the activation of epithelial markers and preventing cells from entering EMT.

Chromatin remodeling is also linked to some of the genomic mechanisms driving carcinoma cell plasticity, and these also lead to invasion and metastasis [26,27]. Carcinoma cells are rarely homogenous, exhibiting multiple chromatin states and EMT states at the same tumorigenic stage. Several methods have been established to help define distinct EMT states and different cell lineages within the same tumor. One approach uses RNA-sequencing or microarray gene expression data to generate an EMT score to define the EMT state [28]. Single-cell barcoding RNA-sequencing has also been established to compute EMT scores from EMT-related gene expression [29]. Cell surface markers EpCAM, CD106, CD51 and CD61 can be used to distinguish among six different EMT states in carcinoma cells that span the EMT spectrum. Common core motifs including AP1, Ets, TEAD and Runx motifs are involved in chromatin remodeling irrespective of whether it leads to the acquisition of EMT or MET states. In contrast, specific transcription factors P63, Zeb1, CTCF and Mafk were specifically associated with chromatin remodeling to adopt different EMT states [30]. These assays show that distinct differentiation levels are associated with particular chromatin landscapes and their regulation of epithelial and mesenchymal marker expression.

## 3. Carcinoma Cell Plasticity Contributes to Therapy Evasion and Stemness

Carcinoma cells that acquire a mesenchymal-like phenotype through EMT are more resistant to cell death [31], and numerous studies have shown a correlation between EMT and chemotherapy resistance [32,33]. For example, in lung carcinoma, EMT can elicit resistance to newly developed targeted therapeutics such as alectinib, ceritinib, and lorlatinib in lung tumors associated with ALK translocations, suggesting EMT as a mechanism leading to drug resistance [34]. Another example is the appearance of refractory carcinoma cells, designated “persister cells”, which acquire a highly mesenchymal cell state in part following the activation of a lipid peroxidase pathway [35].

Plasticity related chemoresistance in carcinoma cells may also be accompanied by transdifferentiation. Non-small cell lung carcinoma (NSCLC) can transdifferentiate into small cell lung carcinoma (SCLC) during the acquisition of resistance to tyrosine kinase inhibitors against epidermal growth factor receptors (EGFRs); albeit, such transdifferentiation has also been noted among patients not exposed to such inhibitors [36]. Neuroendocrine transdifferentiation occurs frequently in castration-refractory prostate carcinoma. Such carcinoma cells derive from prostate adenocarcinoma cells rather than from the neuroendocrine cells present in the gland. These cells undergo a progressive transition through intermediate stages, acquiring a plastic phenotype and finally losing the expression of their androgen receptors [37]. Thus, plastic carcinoma cells may evade chemotherapy or targeted therapeutics by augmenting cell plasticity through more extensive EMT pathways, and possibly accompanied by transdifferentiation. The resistance to drugs will inevitably lead to tumor recurrence, making the development of new drugs able to overcome this refractoriness hard to design. These findings prompt the inception of studies that can unravel the mechanisms driving carcinoma cell plasticity, with the ultimate goal of identifying druggable targets that can interfere with the acquisition of plasticity in carcinoma cells.

Immune evasion is another important yet poorly understood aspect associated with carcinoma plasticity. The tumor microenvironment (TME) may play a role in the association between carcinoma cell plasticity and immune evasion. In addition, the acquisition of a mesenchymal-like phenotype in carcinoma cells could be a mechanism that allows cells to escape immune surveillance and resist immunotherapy. It is now well established that TGF-β is not only an EMT inducer in carcinoma, but is also a potent driver of immune evasion [38]. TGF-β can induce the upregulation of PD-L1 through Smad2 phosphorylation to promote immune suppression in NSCLCs. Treatment with the bifunctional agent bintrafusp-α, which targets both PD-L1 and TGF-β, could decrease the expression of mesenchymal genes and suppress proliferation rates, both in vivo and in vitro [39]. TGF-β also inhibits T cell proliferation by decreasing IL-2 through the cooperation of Smad3, Smad4 and the co-factor TOB1 [40,41]. Likewise, TGF-β can inhibit naïve T cells from differentiating into T_H_1 (type 1 T-helper) cells, which are responsible for the T cell response against carcinoma cells [42,43,44]. Tumor organoids were generated by crossing genetically modified mice carrying four key human CRC mutations: Apc^fl/fl^, Kras^LSL-G12D^, Tgfbr2^fl/fl^, and Trp53^fl/fl^ (AKTP) [45]. Ninety percent of mice bearing the quadruple AKTP mutation developed CRC, and over half progressed to the metastatic T4 stage. Stromal cells localized at the invasive margins expressed high levels of TGF-β, which was associated with immune evasion, as indicated by the expression of CALD1 and IGFBP7, two TGF-β–induced genes. Overexpression of TGF-β by carcinoma-associated fibroblasts (CAFs) can promote the exclusion of CD4+ and CD8+ T cells from the invasive carcinoma. Treatment with the TGF-β inhibitor, galunisertib, in the mouse quadruple AKTP model was able to enhance the T cell response against the carcinoma cells and prevent metastasis. Finally, overexpression of interleukin-8 (IL-8) is observed in various types of carcinomas and in tumor-associated macrophages, suggestive of a close relationship between IL-8 and the tumor microenvironment [46]. In tumor cells, IL-8 signaling can promote EMT and stemness, which drive invasiveness and metastasis [47]. Altogether, tumor plasticity driven by EMT could contribute to immune evasion and to immunotherapy resistance.

The concept of the “carcinoma stem cell” (CSC) was postulated decades ago in studies on teratocarcinoma [48]; this field was later revigorated by studies focusing on leukemia [49]. However, aside from teratocarcinoma, it has been challenging to isolate pure populations of CSCs from solid tumors. An important issue is that CSCs may exhibit high plasticity such that stemness may be a stochastic process. Normal stem cell self-renewal occurs through asymmetric cell division, with one daughter cell permitted to proliferate and differentiate to form one of the diverse cell phenotypes of the bulk of the tumor, while the other daughter cell maintains the stem-like features of the parent cell for further stem cell division. However, these processes are rarely identified in enriched fractions of CSCs, which are obtained by sorting carcinoma cells based on CSC markers (e.g., ALDH1, CD44, CD90 CD133, CD166 in lung carcinoma). Thus, the use of tumor initiating cells (TICs) or tumor propagating cells (TPCs) is preferable to CSC, as these terms are operational terms [50]. CSCs were initially found to be enriched in advanced breast carcinoma exhibiting a CD44^+^CD24^–/low^ phenotype [51]. However, there are potentially distinct stem cells in breast cancers harboring different EMT phenotypes [52]; for example, in the skin, the IFE and HF harbor distinct stem cells. During normal homeostasis, lineage-specific transcription factors KLF5 and SOX9 are specifically expressed in IFE epidermal stem cells (EdSCs) and HF stem cells (HFSCs), respectively [53]. However, during wound-repair and tumor progression, these two TFs are co-expressed along with activation of the stress-TFs, ETS2 and STAT3, which leads to lineage infidelity. Loss of either KLF5 or SOX9 inhibits SCC tumor progression in nude mice. Only SCCs derived from HF engage in EMT in xenografts, which suggests that transcriptional priming of EMT is likely to be cell-lineage dependent [54].

TICs and TPCs in EMT-induced quiescence are highly resistant to chemotherapy, presumably because these carcinoma cells are highly plastic [55,56,57]. For instance, NOTCH4, which is highly expressed in triple-negative breast carcinoma (TNBC), contributes to EMT and maintains the cells in a mesenchymal-like state through an upregulation of SNAI2 and GAS1 [58]. These cells become quiescent and demonstrate enhanced stemness along with chemoresistance. In CRC, chemoresistance was observed in a group of quiescent cells. These quiescent cells expressed high levels of ZEB2, which led to an increase in the expression of pCRAF/pASK1 and resulted in EMT and chemoresistance. Indeed, in patients with CRC, elevated ZEB2 is often associated with a high expression of EMT genes, decreased proliferative rates, and poor prognosis [59].

To overcome some of the issues associated with identifying CSCs, one approach is to label CSCs with stem cell-specific markers using inducible genomics. As an example, GFP was inserted into the gene encoding for the leucine rich repeat-containing G-protein–coupled receptor 5 (*LGR5*) stem marker using the CRISPR-Cas9 system [60]. LGR5-GFP human colorectal carcinoma (CRC) cells as well as KRT20 (keratin 20)-GFP CRC cells were generated and permitted the formation of CRC organoids (COOs) in Matrigel [61]. The LGR5-GFP COOs expressed high levels of stem cell genes (*SMOC2*, *RGMB* and *LRIG1*) whereas KRT20-GFP COOs expressed genes related to differentiation. The mRNA levels of LGR5 and KRT20 in xenografts derived from the LGR5-GFP COOs resembled the expression patterns found in the original patient samples, as detected using in situ hybridization. Notably, LGR5–COOs can regenerate LGR5+ cells in the secondary culture, indicating the plasticity of the cells to resume a stemness state. Spheroid-forming capacity can be regarded as an evaluation of stemness and it is enhanced by overexpression of the EMT-TFs SNAI1, SNAI2, TWIST, and ZEB1 with the induction of stem cell-related gene expression [62,63,64,65].

CSCs are highly plastic; yet, it is still debated whether plasticity resulting from EMT will directly lead to stemness and induce metastasis [66,67]. Recently, 3D spheroids of breast carcinoma cells were found to express epithelial and mesenchymal markers, revealing a “hybrid EMT state.” Hybrid EMT states have the potential to re-distribute in a wider range of EMT scores, thus exhibiting more plasticity, and consequently, more efficiently gaining stemness and metastatic potential [30,68,69]. Single-cell RNA sequencing of TGF-β-induced breast cell line MCF10A showed that EMT signaling pathways are activated sequentially after induction, while stemness associated pathways are more readily activated. Moreover, breast cancer patients whose transcriptomes exhibit hybrid EMT signatures have a lower survival rate [70]. This finding supports that hybrid EMT states as compared to mesenchymal states, may enable carcinoma cells to be more adaptive to therapies and to acquire refractoriness to these therapies. Intermediate EMT states are thus less vulnerable and more adaptive to execute all steps of the metastatic cascade. However, there is still a debate as to whether and to what extent partial or hybrid EMT is required for metastasis [71,72,73,74,75]. The main issues lie in the murine models used and the mesenchymal reporter FSP1, which does not faithfully capture the full-blown mesenchymal phenotype. Others have shown that deletion of *Fat1* in SCCs promotes an EMT hybrid state and augments cellular stemness and metastasis [76]. Thus, EMT plasticity status may be a critical parameter in studying the acquisition of stemness.

## 4. TGF-β Signaling in Carcinoma Plasticity

The transforming growth factor β (TGF-β) family is a large family of evolutionarily conserved molecules, including TGF-β1, TGF-β2, activins, inhibins, nodal, Mullerian-inhibiting substance (MIS), growth and differentiation factor (GDF), and bone morphogenetic proteins (BMPs). TGF-β family signaling is activated by the binding of ligands to type I and two type II cell-surface-specific receptor complexes. The binding of TGF-β1/2 to their cognate type II receptors induces type II receptor phosphorylation and its subsequent binding to the type I receptors. Activated type I receptors then phosphorylate the Smad2/3 signal transducers at the carboxy-terminal domain. The regulatory Smad2/3 proteins then form a trimeric complex with the co-Smad4 (Smad2/3/4) and translocate into the nucleus to regulate target gene transcription [77].

Pulmonary fibrosis [78], wound-healing [79], and embryonic stem cell (ESC) differentiation [80] are all associated with upregulated TGFβ-1/2 expression. TGF-β plays different roles in regulating cell plasticity; in normal and premalignant cells, TGF-β behaves like a tumor suppressor, whereas in carcinoma, it is involved in promoting oncogenicity [81]. In pre-malignant cells, TGF-β can induce the expression of cyclin-dependent kinase (CDK) inhibitors 1A (CDKN1A or p15), 2B (CDKN2B or p21), 1C (CNKN1C or p57), and death-associated protein kinase (DAPK). This leads to cell cycle arrest, and the cells are forced into apoptosis [82,83,84,85]. However, various carcinoma types, including breast [86], lung [87], pancreatic [88], and colorectal carcinoma [89] show evidence of TGF-β overexpression. In these carcinoma types, high TGF-β expression is often associated with an increased level of EMT, which drives invasiveness, metastasis, and carcinoma stemness [88,90,91,92]. The EMT that is associated with TGF-β/Smads signaling can be suppressed by lncRNA Smad3-associated long non-coding RNA (SMASR), which are close to Smad3 in lung cancer cells [93]. Similarly, in lung cancer cells, TGF-β induced EMT could also be suppressed by quaking 5 (QKI-5) protein via binding directly to the 3′UTR region of TGFR1 to degrade its mRNA [94].

However, mutations in TGF-β pathway downstream effectors can also act to promote cell survival and growth through reduced rates of apoptosis. For example, inactivating mutations in *SMAD*, which can be found in colorectal carcinoma, lung carcinoma, and hepatocarcinoma [95,96], cause a reduction in apoptosis, likely through regulation of E3 ubiquitin ligase [32]. Furthermore, in pancreatic ductal adenocarcinoma, overexpression of TGF-β induces the expression of Snail and Sox4 via the canonical Smad2/3/4 signaling pathway. Smads/Snail will drive cells to undergo EMT by repressing Klf5 expression, which is lethal under sustained Sox4 expression. Conversely, in the absence of Smad4, survival and tumor propagation are ensured in pancreatic carcinoma cells; Snail is repressed instead of Klf5, leading to cooperation between Klf5 and Sox4 to prevent apoptosis [97].

The induction of EMT by TGF-β allows cells to gain plasticity, stemness, and refractoriness to chemo- and targeted therapies, and this aspect of TGF-β signaling remains to be explored in depth. Many studies have focused on the dichotomous effect of TGF-β in its switch from a tumor suppressor to a tumor promoter; such studies could deliver potential targets for cancer treatment. RUNX3—the gene that encodes for runt-related transcription factor 3—is a known tumor suppressor and is associated with inhibiting the pro-oncogenic effect of TGF-β [98]. RUNX3 prevents the accumulation of reactive oxygen species (ROS) via upregulation of redox regulator heme oxygenase-1 (*HMOX1*). ROS is part of the redox signaling pathway, and induces oxidative DNA damage. Loss of RUNX3 will enhance TGF-β-mediated oncogenesis in lung carcinoma cells, along with an increase in ROS mediated by the TGF-β pathway.

Paraspeckle component 1 (PSPC1) has also been identified as a contextual determinant of TGF-β switching [99]. PSPC1 expression profiling, derived from The Cancer Genome Atlas (TCGA), shows a correlation between overexpression and poorer patient survival. Moreover, in carcinoma cells, upregulation of PSPC1 can increase TGF-β autocrine signaling, and can directly interact with phosphorylated Smads in the nucleus to activate the expression of EMT-related genes, thereby leading to EMT and stemness. In PSPC1-deficient cells, TGF-β could act as a tumor suppressor by activating cell-cycle arrest genes like *p15*, *p21*, *p57* and *DAPK1*. Besides RUNX3 and PSPC1, there are reports of other factors that could modulate this dichotomous TGF-β signaling; these are listed in Figure 1. These factors could help carcinoma cells escape the tumor suppressive effect of TGF-β and switch TGF-β into a tumor promoter.

A recent study suggests that TGF-β can promote a reversible cell-cycle arrest that prevents squamous cell carcinoma (SCC) cells from responding to chemotherapy. Using an inducible histone H2B-GFP system, the authors identified a subset of GFP-retaining cells from among SCC tumors that expressed low levels of the transferrin receptor CD71, had acquired a quiescent state, and had become resistant to chemotherapy [55]. In these quiescent cells, G1 arrest was reversible, and the cells could re-acquire a proliferative status, regulated by Smad2-dependent transcripts and inactivation of TGFβRII. These findings suggest that TGF-β can promote a reversible cell-cycle arrest to prevent SCCs from responding to chemotherapy in tumors.

Chemotherapy tends to target only proliferative carcinoma cells, which allows quiescent carcinoma cells to survive and become the “cell of origin” in tumor relapse; such relapse is usually associated with poor prognosis. Tumor-propagating cells (TPC) share features with quiescent cells; they are considered to be less proliferative but exhibit plasticity. To further understand the origin of such tumor-propagating cells in SCC, one study isolated SCCs derived from the interfollicular epidermis (IFE) and hair follicle (HF) [54]. IFE-derived SCCs retained an epithelial status and a high expression of EpCAM, whereas HF-derived SCCs exhibited EMT through TGFβR activation with a gradual loss of EpCAM expression; these findings suggest that a pre-EMT state is acquired by the cells of origin in HF. In these cells of origin, transcriptional priming of EMT is established early as a key mechanism to confer plasticity to tumor-initiating cells (TICs). IFE-derived SCCs have a sustained expression of p63, which renders the cells resistant to TGF-β-induced EMT. This may suggest that sustained p63 expression protects these cells from undergoing EMT. One intriguing issue is whether such primed cells are already present in healthy or pre-malignant epidermal cells as a key factor that engenders the refractoriness of tumor cells to chemotherapies. Other studies also prove that TGF-β-induced EMT is associated with drug resistance in carcinoma [104,105,106]. Thus, targeting TGF-β-related EMT in carcinoma could help to improve the drug efficiency and partially solve resistance issues.

## 5. Therapeutic Potential of Inhibiting TGF-β-Related EMT in Carcinoma

We have shown that carcinoma cell plasticity can promote therapeutic resistance. However, how the EMT status confers chemoresistance and immune evasion remains elusive. Many studies have focused on developing drugs to abrogate plasticity or to prevent cells from acquiring a plastic state. Such drugs can target EMT signaling pathways driven by EGF, Notch, ERK5, NF-κB and PI3K; by transcription factors such as TWIST and co-factors like HDAC1/2 and by specific metabolic routes [107,108,109]. One of the therapeutic targets for diminishing EMT in carcinoma cells is TGF-β signaling, since it is a major inducer of EMT (Figure 2). In malignant tumors, TGF-β is produced by the tumor microenvironment, particularly by CAFs, leading to the activation of EMT in neighboring carcinoma cells, inactivation of T_H_1 cell differentiation, and immune surveillance escape [110]. Minnelide, a water-soluble compound modified from a compound extracted from the Chinese plant *Tripterygium wilfordii* was discovered to reduce the viability of CAFs taken from pancreatic tumors [111], and thus, reduce TGF-β secretion [112]. Minnelide is under phase II clinical study to evaluate its effect on relapse pancreatic tumors (NCT03117920) along with other phase I clinical studies on advanced gastric carcinoma (NCT03129139). ABBV-151, a first-in-class monoclonal antibody (mAb), can block TGF-β1, TGF-β2, TGF-β3, and glycoprotein A repetitions predominant (GARP) to reverse immune suppression [113]; this antibody is also in phase I clinical trials against advanced solid tumors (NCT03821935). Various preclinical studies have explored the potential anti-tumor effect of antibodies that block systemic TGF-β [114]. NIS793 and SAR439459, two broad anti-TGF-β antibodies, have been developed and used alongside anti-PD1 antibodies as a combinatorial therapy to enhance the inhibition of tumor growth and metastasis. Both of these antibodies are in phase I clinical trials for the treatment of advanced malignant tumors (NCT02947165 and NCT03192345, respectively). Several humanized antibodies, such as fresolimumab and TβM1, have also been developed and are in clinical trials. [115,116]. In particular, fresolimumab has been shown to increase the survival of patients with metastatic breast carcinoma when delivered in combination with focal radiotherapy [116]. An isoform selective inhibitor of only TGF-β1/3, AVID200, is also in phase I clinical trials for the treatment of malignant solid tumors (NCT03834662).

The most widely applied strategy to inhibit TGF-β signaling is to target TGFR kinases using small molecules. Among the small molecules that have been tested, galunisertib (LY2157299) is in clinical trials for the treatment of advanced solid tumors, including hepatocellular, pancreatic, and prostate carcinoma. Vactosertib (TEW-7197), another small molecule compound recently developed to specifically target the adenosine-5-triphosphate (ATP) binding site of TGFβRI, is in phase I and phase II clinical trials for the treatment of various carcinomas in combination with chemotherapy or immune checkpoint antibodies. Besides these, several other compounds (LY3200882 and PF-06952229) are also being tested for their inhibitory action against TGFR kinase activity. LY3022859, a monoclonal antibody against TGFβRII, can inhibit TGFβRII-mediated signaling and is also under phase I clinical trials against solid tumors [117]. Bintrafusp-α, a bifunctional protein, is a PD-L1 antibody fused with the extracellular domain of the human TGFβRII, which acts to trap TGFβ while also blocking PD-L1. Bintrafusp-α is currently being tested in over 30 active phase I trials against multiple carcinoma types.

Vaccination could be a potentially powerful therapeutic approach since vaccines are produced specifically to stimulate the immune response against certain types of carcinomas. Belagenpumatucel-L is an advanced tumor vaccine that uses four irradiated and allogeneic NSCLC lines (large cell carcinoma, squamous cell carcinoma and adenocarcinoma) transfected with a plasmid containing the TGF-β2 antisense transgene to stimulate an immune response. Belagenpumatucel-L has been studied in clinical trials in NSCLC with good tolerance and offers a survival advantage [118]. Vigil (gemogenovatucel-T), using the same principle, is a vaccine based on the ex vivo transfection of autologous carcinoma cells with a short hairpin RNAi combined with the GM-CSF transgene targeting furin to downregulate TGF-β1 and TGF-β2 proteins. Clinical trials with Vigil also demonstrate a decrease in recurrence and improvement of survival rate, especially in ovarian cancer patients [119].

Given the role of TGF-β in the emergence of therapeutic resistance, the development of therapeutic approaches that target it and downstream signaling could lead to more durable responses. As described above, strategies are currently developed to inhibit cell plasticity driven by TGF-β. Combining targeted therapeutics with chemotherapy and immunotherapy could result in higher efficiency to eliminate residual tumor cells.

## 6. Conclusions and Perspectives

Although it is an essential process in embryonic development, epithelial cell plasticity also contributes to tumorigenesis and metastasis, and despite a significant focus on understanding the mechanisms governing epithelial cell plasticity in recent years, the molecular and cellular machineries involved in this process require further investigation. It is noteworthy that EMT is not the sole cause of therapy resistance. Other mechanisms include autophagy and the acquisition of oncogenic mutations such as Kirsten rat sarcoma viral oncogene homolog (KRAS) [120,121,122]. Autophagy can maintain genomic stability by eliminating damaged cellular components, however, it could drive the same effect to help cancer cells increase resistance to cancer drugs. KRAS mutation was found in over 30% of cancer patients and was thought to be “undruggable” until the KRAS^G12C^ mutation was discovered with a new targeting pocket in the switch II domain of KRAS [123,124]. However, KRAS secondary mutations can also result in new resistance to KRAS^G12C^ inhibitors that require further studies in this field [125].

This review discusses the recent findings that exemplify the complexity of epithelial cell plasticity, with a particular focus on EMT-TFs, chromatin remodeling, stemness, resistance to therapeutics including immune avoidance and TGF-β–related regulation and therapeutic approaches. Clearly, the many questions still outstanding—despite the broad scope of the information—point to the lack of a detailed understanding of the entangled mechanisms. For example, what degree of cell plasticity could affect carcinoma invasion and metastasis? What are the mechanisms operating under a plasticity background driving chemo- and targeted therapy resistance? Which factors in the TGF-β signaling pathway contribute to cancer cell plasticity? How to identify novel potential targets in the TGF-β signaling pathway to combat carcinoma plasticity-related chemoresistance and immune evasion? Recent advances in single-cell sequencing, high-resolution lineage tracing, and methods that seek to identify chromatin landscapes related to gene regulatory networks could help to answer some of these questions. These approaches may help to clarify the properties of different intermediate states of EMT and to further assess the influence of TGF-β signaling pathway in carcinoma plasticity. Thus, developing methods to better qualify cell plasticity stages and decode the relationship between TGF-β-associated carcinoma plasticity and drug resistance is an essential step for the discovery of novel potential targets in the progression of carcinoma.

## Figures and Tables

**Figure 1 cancers-13-03397-f001:**
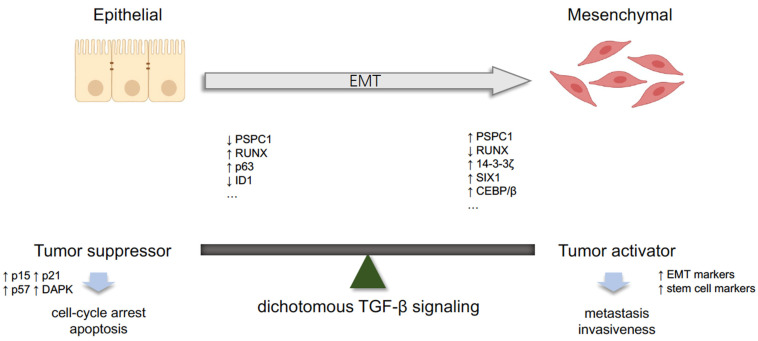
Dichotomous transforming growth factor beta (TGF-β) signaling pathway. In normal cells or early malignant cells, TGF-β works as a tumor suppressor. Downregulation of PSPC1 [99], RUNX [98], p63 [55] and ID1 [100] could be associated with cell-cycle arrest and apoptosis. In advanced tumors, TGF-β acts as a tumor activator with upregulation of PSPC1 [99], RUNX [98], 14-3-3ξ [101], SIX1 [102] and CEBP/β [103]. Upward arrow represents upregulation of indicated factors and downward arrow represents downregulation of indicated factors.

**Figure 2 cancers-13-03397-f002:**
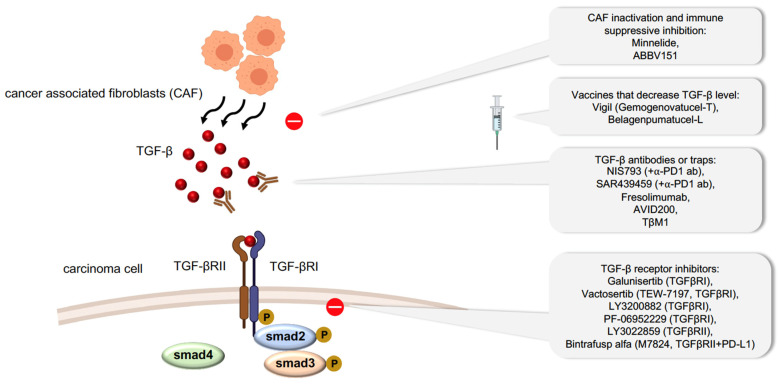
Therapeutic approaches targeting TGF-β signaling pathway.

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
