# Peer review of "Harnessing Carcinoma Cell Plasticity Mediated by TGF-β Signaling"

_cancers, 2021, doi:10.3390/cancers13143397_

Round 1

Reviewer 1 Report

This review focuses on cancer cell plasticity, with a specific highlight on the role of TGFb signalling, EMT transcription factors and chromatin remodelling. This is timely review on a critical process in the cancer field. In one hand, the field is very rich but at the same time, it is a very fuzzy and clearly needs good reviews clarifying the cancer cell plasticity concept. Yet, I found the text not well constructed. To my point of view, the section on “Cancer cell plasticity” should include the section “Carcinoma stem cells” where a discussion on the link between both and with metastasis, specially that cancer initiating cells is used in the text before introducing this section. This part should also include the notion of intratumoral heterogeneity and the observations that carcinoma contains cells with different EMT states (line 147), also Tumour-propagating cells (p 222) and chemoresistance (which is also referred to the text also before the specific section line 325) and immune invasion. When those concepts are clearly introduced and related to each other, the authors could go to molecular mechanisms involving TGFb signalling. And therapeutic potential of inhibiting TGFb signalling. As a whole I would suggest the authors to make TGFb signalling and cancer cell plasticity their focus, as the way it stands, the text is messy, leaving the impression that we have no idea where the authors wants to take us.

Below are also some specific points:

The title should be more specific to me with a mention to TGFb.

Line 8: The authors state: “Carcinoma cell plasticity is achieved through epithelial-mesenchymal transition (EMT), with cells positioned seemingly indiscriminately across the spectrum of EMT phenotypes.” The authors exclude data showing that hybrid EMT phenotypes could more plastic than the pure epithelial or mesenchymal phenotypes.

Line 21: The authors state: “In carcinoma, plasticity is an exquisite adaptive mechanism to respond to stressors in the microenvironment, endowing cells not only with the capacity to avoid apoptosis but also to acquire stemness and refractoriness to therapy.” I would give here a more complete definition on cancer cell plasticity, which also include the cell ability to reprogramme into other cell linages, as mentioned in line 61.

Line 47: “The key mechanism driving carcinoma cell plasticity is epithelial-mesenchymal transition (EMT):….” I would be careful here saying: A key mechanism driving …..Can we really state at this stage that the activation of an EMT programme is the only way by which cells acquire plastic behaviour? Is transdifferentiation associated with EMT? EMT should also be defined more carefully here, including the notion of hybrid phenotypes, as referred line 282.

Line 77: The title “Transcriptional regulation of carcinoma plasticity” is not well adapted to me. TGFb also induces cancer cell plasticity through changes in transcriptional regulation. I did not really understand why the authors focus more on EMT transcription factors and chromatin remodelling. If so, they should discuss the questions related to this at the end and relates this to TGFb signalling.

Line 142: “Several methods have been established to help define distinct EMT states and different cell lineages within the same tumor. One approach uses RNAsequencing or microarray gene expression data to generate an EMT score to define the EMT state [34]. Single-cell barcoding RNA-sequencing has also been established to compute EMT scores from EMT-related gene expression [35]. Cell surface markers EpCAM/CD106/CD51/CD61 can be used to distinguish among six different EMT states in carcinoma cells that span the EMT spectrum [36]. These assays show that distinct differentiation levels are associated with particular chromatin landscapes and their regulation of epithelial and mesenchymal marker expression.” I do not see data described here showing that distinct differentiation levels are associated with particular chromatin landscapes. The ref 36 shows such data but the authors should include what are these data.

Section on TGFb: The authors include detailed data on the tumour suppressor role of TGFb. This is not the place to me, as the review focuses on cancer cell plasticity. On sentence reffering to that would do. This also takes away the focus of the reader. Instead, the authors could enrich this part by providing a more complete literature on TGFb and EMT/cell plasticity/therapy resistance.

Line 293: “ These quiescent cells expressed high levels of ZEB2, which led to an increase in the expression of pCRAF/pASK1 and resulted in EMT and chemoresistance. Indeed, in patients with CRC, elevated ZEB2 is often associated with a high expression of EMT genes, decreased proliferative rates, and poor prognosis.” Why this does not go to the part on transcriptional regulation”?

Line 330: “In malignant tumors, TGF- is produced by the tumor microenvironment, particularly by CAFs, leading to the activation of EMT in neighboring carcinoma cells, inactivation of TH1 cell differentiation, and immune surveillance escape [96].” This should not go to the therapeutic potential section.

Conclusion and Perspectives: The authors ask good questions related to cancer cell plasticity but what I was expecting here were specific questions on TGFb signalling and cancer cell plasticity and on transcription factors/chromatin remodelilng if the authors really want to keep this part.  

Author Response

Reviewer 1

This review focuses on cancer cell plasticity, with a specific highlight on the role of TGFb signalling, EMT transcription factors and chromatin remodelling. This is timely review on a critical process in the cancer field. In one hand, the field is very rich but at the same time, it is a very fuzzy and clearly needs good reviews clarifying the cancer cell plasticity concept. Yet, I found the text not well constructed. To my point of view, the section on “Cancer cell plasticity” should include the section “Carcinoma stem cells” where a discussion on the link between both and with metastasis, specially that cancer initiating cells is used in the text before introducing this section. This part should also include the notion of intratumoral heterogeneity and the observations that carcinoma contains cells with different EMT states (line 147), also Tumour-propagating cells (p 222) and chemoresistance (which is also referred to the text also before the specific section line 325) and immune invasion. When those concepts are clearly introduced and related to each other, the authors could go to molecular mechanisms involving TGFb signalling. And therapeutic potential of inhibiting TGFb signalling. As a whole I would suggest the authors to make TGFb signalling and cancer cell plasticity their focus, as the way it stands, the text is messy, leaving the impression that we have no idea where the authors wants to take us.

We very much thank the reviewer for suggested improvements on the manuscript. We moved the TGF-beta signalling related to plasticity section next to the paragraph dealing with targeting TGF-beta to focus on the relationship between TGF-beta signalling and carcinoma plasticity. We also introduce more studies focusing on partial EMT and describe more precisely transcriptional regulation involved in chromatin remodelling along the EMT spectrum.

Below are also some specific points:

The title should be more specific to me with a mention to TGFb.

We modified the title accordingly

Line 8: The authors state: “Carcinoma cell plasticity is achieved through epithelial-mesenchymal transition (EMT), with cells positioned seemingly indiscriminately across the spectrum of EMT phenotypes.” The authors exclude data showing that hybrid EMT phenotypes could more plastic than the pure epithelial or mesenchymal phenotypes.

We added several articles to better introduce the concept of hybrid EMT phenotypes.

Line 21: The authors state: “In carcinoma, plasticity is an exquisite adaptive mechanism to respond to stressors in the microenvironment, endowing cells not only with the capacity to avoid apoptosis but also to acquire stemness and refractoriness to therapy.” I would give here a more complete definition on cancer cell plasticity, which also include the cell ability to reprogramme into other cell linages, as mentioned in line 61.

We included this important point in the introduction.

Line 47: “The key mechanism driving carcinoma cell plasticity is epithelial-mesenchymal transition (EMT):….” I would be careful here saying: A key mechanism driving …..Can we really state at this stage that the activation of an EMT programme is the only way by which cells acquire plastic behaviour? Is transdifferentiation associated with EMT? EMT should also be defined more carefully here, including the notion of hybrid phenotypes, as referred line 282.

We changed this sentence into “one of the key mechanisms…”

Line 77: The title “Transcriptional regulation of carcinoma plasticity” is not well adapted to me. TGFb also induces cancer cell plasticity through changes in transcriptional regulation. I did not really understand why the authors focus more on EMT transcription factors and chromatin remodelling. If so, they should discuss the questions related to this at the end and relates this to TGFb signalling.

We wanted to relate the EMT-TFs to their potential functions in chromatin remodeling since this field is relatively novel in EMT research. We cited recent studies related to this expanding field to provide readers with another facet of how EMT can be regulated.

Line 142: “Several methods have been established to help define distinct EMT states and different cell lineages within the same tumor. One approach uses RNAsequencing or microarray gene expression data to generate an EMT score to define the EMT state [34]. Single-cell barcoding RNA-sequencing has also been established to compute EMT scores from EMT-related gene expression [35]. Cell surface markers EpCAM/CD106/CD51/CD61 can be used to distinguish among six different EMT states in carcinoma cells that span the EMT spectrum [36]. These assays show that distinct differentiation levels are associated with particular chromatin landscapes and their regulation of epithelial and mesenchymal marker expression.” I do not see data described here showing that distinct differentiation levels are associated with particular chromatin landscapes. The ref 36 shows such data but the authors should include what are these data.

We added more information especially chromatin landscapes associated with the different positions in the EMT spectrum.

Section on TGFb: The authors include detailed data on the tumour suppressor role of TGFb. This is not the place to me, as the review focuses on cancer cell plasticity. On sentence reffering to that would do. This also takes away the focus of the reader. Instead, the authors could enrich this part by providing a more complete literature on TGFb and EMT/cell plasticity/therapy resistance.

We thought that dichotomous TGF-β signalling pathway discussed in this section points a link to plasticity and as such it was sufficient to exemplify how TGF- β influences the plasticity of carcinoma cells by either suppressing or activating EMT. We have added more references in this section.

Line 293: “ These quiescent cells expressed high levels of ZEB2, which led to an increase in the expression of pCRAF/pASK1 and resulted in EMT and chemoresistance. Indeed, in patients with CRC, elevated ZEB2 is often associated with a high expression of EMT genes, decreased proliferative rates, and poor prognosis.” Why this does not go to the part on transcriptional regulation”?

We thought that this quiescent cell state is more related to chemoresistance prompting us to leave it in this section 

Line 330: “In malignant tumors, TGF-b is produced by the tumor microenvironment, particularly by CAFs, leading to the activation of EMT in neighboring carcinoma cells, inactivation of TH1 cell differentiation, and immune surveillance escape [96].” This should not go to the therapeutic potential section.

We wanted to focus on how to interfere with TGF-β production in the TME with  specific therapeutics

Conclusion and Perspectives: The authors ask good questions related to cancer cell plasticity but what I was expecting here were specific questions on TGFb signalling and cancer cell plasticity and on transcription factors/chromatin remodelilng if the authors really want to keep this part.  

Chromatin remodelling in TGF-b-induced EMT is still fragmentary,  thus we could not elaborate as yet on this approach     

Reviewer 2 Report

The authors generated a well written review on cancer plasticity, with a specific focus on EMT and TGFb signaling. 

Comments:

  1. The review is well written and easy to understand. The introduction paragraphs were particularly engaging.
  2. As a whole, the title, "Harnessing Carcinoma Cell Plasticity" is fairly misleading. The review takes a rather narrow view of cell plasticity and even within the subject of EMT, the authors do not discuss MET or partial EMT (both of which significantly contribute to metastasis). Given that there are many reviews on EMT and TGFb signaling, it would be a more impactful publication to discuss plasticity as a broad concept that expands beyond what is currently presented. 
  3. Similarly, a more expanded view of how we can combat cellular plasticity using therapeutics (not just TGFb inhibitors) would increase the significance of the article.
  4. A paragraph on signaling plasticity, state change, and drug induced resistance outside of EMT should be added. As well as EMT induced in response to therapeutics
  5. A paragraph on epigenetic marks and chromatin modifiers should be included in transcriptional regulation section
  6. A table of the specific clinical trial compounds and their associated results should be included

Author Response

Reviewer 2

The authors generated a well written review on cancer plasticity, with a specific focus on EMT and TGFb signaling. 

Comments:

  1. The review is well written and easy to understand. The introduction paragraphs were particularly engaging.
  2. As a whole, the title, "Harnessing Carcinoma Cell Plasticity" is fairly misleading. The review takes a rather narrow view of cell plasticity and even within the subject of EMT, the authors do not discuss MET or partial EMT (both of which significantly contribute to metastasis). Given that there are many reviews on EMT and TGFb signaling, it would be a more impactful publication to discuss plasticity as a broad concept that expands beyond what is currently presented. 

We thank the reviewer to point out that the title is misleading.  We have changed it to better fit with the content of the review. We intended to focus on carcinoma cell plasticity, and this is why we did not discuss MET albeit it is an important point which needs to be covered in another review.

  1. Similarly, a more expanded view of how we can combat cellular plasticity using therapeutics (not just TGFb inhibitors) would increase the significance of the article.

In this review, we included recent findings on therapeutic targets related to TGF-β signalling pathway. We have now added some recently published reviews which described more extensively potential therapeutics targeting EMT. We hope that the readers will benefit from these reviews.

  1. A paragraph on signaling plasticity, state change, and drug induced resistance outside of EMT should be added. As well as EMT induced in response to therapeutics

 We have now expanded the text on partial EMT and drug resistance caused by mechanisms unrelated to EMT.

  1. A paragraph on epigenetic marks and chromatin modifiers should be included in transcriptional regulation section

We  added more papers on chromatin remodelling

  1. A table of the specific clinical trial compounds and their associated results should be included

We thought the last paragraph contains this information.

Reviewer 3 Report

Xuecong and Thiery detail a very nice review of tumor cell plasticity with a specific focus on how TGFB drives the acquisition of this phenotypic plasticity and promotes therapy resistance.  While this is a TGFB focused review, overall aspects of tumor cell plasticity were included, which is very helpful to the reader. The manuscript is very well written and easy to follow. 

Minor point:

1. M7824 and Bintrafusp alfa are mentioned at different locations in the text as seemingly separate drugs. In reality, these agents are simply the same drug with different naming.  It would be less confusing to readers if the naming was unified maybe as Bintrafusp alfa (M7824).  Also, citations for Bintrafusp alfa would be helpful.

Author Response

Reviewer 3

Xuecong and Thiery detail a very nice review of tumor cell plasticity with a specific focus on how TGFB drives the acquisition of this phenotypic plasticity and promotes therapy resistance.  While this is a TGFB focused review, overall aspects of tumor cell plasticity were included, which is very helpful to the reader. The manuscript is very well written and easy to follow. 

Minor point:

  1. M7824 and Bintrafusp alfa are mentioned at different locations in the text as seemingly separate drugs. In reality, these agents are simply the same drug with different naming.  It would be less confusing to readers if the naming was unified maybe as Bintrafusp alfa (M7824).  Also, citations for Bintrafusp alfa would be helpful.

We thank the reviewer for his supportive words. We changed M7824 into Bintrafusp.

Reviewer 4 Report

The review Xuecong and Thiery provides an update of recent advances regarding the impact of cell plasticity, and the epithelial-to-mesenchymal transition (EMT) process, on tumor-associated phenotypes including stem cell characteristics and therapy resistance. The text is logically organized and clearly written, and the authors provide timely updates on the role of cellular plasticity on stem cell properties, therapy resistance and targeting EMT (and TGFb as an important EMT inducer) as a therapeutic strategy. I have included some suggestions regarding additional content that the authors might consider adding in a revised manuscript.

Major points:

1) The authors briefly describe the essential features of an EMT on page 2 of the review article. However, an expanded description of the EMT and MET processes, with reference to the revised guidelines that were released in 2020 (PMID: 32300252) would be useful to highlight for the reader. The idea of a partial EMT could be discussed.

2) Many readers will be familiar with the EMT process in the context of enhancing cell migration, invasion, and metastasis. However, in recent years, several publications have come out that question whether EMT is an obligate step in the metastatic cascade (PMID: 26560033; PMID: 26560028; PMID: 30120146). Inclusion of a section that discusses these differing views, with a brief description of the multiple modes of cancer cell migration, invasion and metastasis will help contextualize the importance of the EMT process, and mesenchymal cell migration, to cancer metastasis

3) The last section focuses on targeting TGFb to block EMT. However, there have been numerous reviews focused on the challenges that are associated with the TGFb pathway as a therapeutic target. Some important context on the caveats/dangers of this approach would be warranted. Also, expanding this section to propose additional targets associated with the EMT process, which can be readily targeted, would be a worthwhile addition (for example: PMID: 31676574).

Author Response

Reviewer 4

The review Xuecong and Thiery provides an update of recent advances regarding the impact of cell plasticity, and the epithelial-to-mesenchymal transition (EMT) process, on tumor-associated phenotypes including stem cell characteristics and therapy resistance. The text is logically organized and clearly written, and the authors provide timely updates on the role of cellular plasticity on stem cell properties, therapy resistance and targeting EMT (and TGFb as an important EMT inducer) as a therapeutic strategy. I have included some suggestions regarding additional content that the authors might consider adding in a revised manuscript.

Major points:

1) The authors briefly describe the essential features of an EMT on page 2 of the review article. However, an expanded description of the EMT and MET processes, with reference to the revised guidelines that were released in 2020 (PMID: 32300252) would be useful to highlight for the reader. The idea of a partial EMT could be discussed.

We thank the reviewer for constructive criticisms. We now discussed more extensively partial EMT.

2) Many readers will be familiar with the EMT process in the context of enhancing cell migration, invasion, and metastasis. However, in recent years, several publications have come out that question whether EMT is an obligate step in the metastatic cascade (PMID: 26560033; PMID: 26560028; PMID: 30120146). Inclusion of a section that discusses these differing views, with a brief description of the multiple modes of cancer cell migration, invasion and metastasis will help contextualize the importance of the EMT process, and mesenchymal cell migration, to cancer metastasis

We now included this issue and discuss the possible reasons that metastasis may be independent of EMT along with partial EMT

3) The last section focuses on targeting TGFb to block EMT. However, there have been numerous reviews focused on the challenges that are associated with the TGFb pathway as a therapeutic target. Some important context on the caveats/dangers of this approach would be warranted. Also, expanding this section to propose additional targets associated with the EMT process, which can be readily targeted, would be a worthwhile addition (for example: PMID: 31676574).

We added more details in this section; we included additional targets and  some excellent recent reviews which provide a relatively exhaustive list of compounds.

Round 2

Reviewer 1 Report

Reading the revised version, the authors do not really answer my comments and I still think that the text is disorganized. The way it stands, the different sections are not well linked, which makes it difficult to have a clear picture on cancer cell plasticity and on why TGFbeta signalling is interesting. I still think that the section 4. Should be integrated in the section 2 with section 5. With a better link on how chemoresistance and immune evasion are associated with cancer cell plasticity. Moreover, it does not make sense to me to describe a substantial paragraph on TGFbeta and immune invasion (line 250-255), and next have a section 6 on TGF-β signaling in carcinoma plasticity. Data on TGFbeta and immune invasion should be integrated in the section TGF-β signaling in carcinoma plasticity, possibly adding sub-sections. I do not understand either why there is a paragraph on chemoresistance (section 5), while this notion is already mentioned in section 1. I also still think that the conclusions and perspectives should focus on TGFb in cancer cell plasticity. The perspectives of trying to understand “Why do carcinoma cells exhibit multiple EMT states in a tumor and how do they co-operate together to promote a more aggressive phenotype? Is a good point but since there is no reference on cooperation in the main text, the question is not related to the current review. The 2 questions: “Which factors in the TGF-b signaling pathway contribute to cancer cell plasticity?” Followed by “What could be the potential targets for developing novel therapeutic strategies against carcinoma plasticity?” make the reader to think that the authors do not believe TGFbeta signalling is an interesting pathway to target.

Author Response

We very much thank the reviewer for further suggestions to improve the manuscript. We followed the recommendations made by the reviewers to merge section 2, 4 and 5. In the conclusion, we added several specific questions related to TGFbeta signalling in carcinoma plasticity and to  therapeutic approaches specifically targeting  this  pathway.

Round 3

Reviewer 1 Report

The last version of the manuscript make more sense in term of organization.